# Harmony: A Joint Self-Supervised and Weakly-Supervised Framework for Learning General Purpose Visual Representations

**Mohammed Baharoon**  *mohammed_baharoon@hms.harvard.edu*
*Harvard University*
*KAUST*

**Jonathan Klein**  *jonathan.klein@kaust.edu.sa*
*KAUST*

**Dominik L. Michels**  *dominik.michels@kaust.edu.sa*
*KAUST*

**Reviewed on OpenReview:** *https://openreview.net/forum?id=IcOBCufqFO*

## Abstract

Vision-language contrastive learning frameworks such as CLIP enable learning representations from natural language supervision and provide strong zero-shot classification capabilities. However, due to the nature of the supervisory signal in these paradigms, they lack the ability to learn localized features, leading to degraded performance on dense prediction tasks such as segmentation and detection. On the other hand, self-supervised learning methods have shown the ability to learn granular representations, complementing the high-level features in vision-language training. In this work, we present Harmony, a framework that combines vision-language training with discriminative and generative self-supervision to learn visual features that can be generalized across different downstream vision tasks. Our framework is specifically designed to work on web-scraped data by not relying on negative examples in the self-supervised learning path and addressing the one-to-one correspondence issue using soft CLIP targets generated by an EMA model. Moreover, Harmony optimizes for five different objectives simultaneously, efficiently utilizing the supervision in each data example, making it even more suited in data-constrained settings. We comprehensively evaluate Harmony across various vision downstream tasks and find that it significantly outperforms the baseline CLIP and outperforms the previously leading joint self- and weakly supervised methods, SLIP, MaskCLIP, and DetailCLIP. Specifically, when compared against these methods, Harmony shows superior performance in linear-probing, fine-tuning, and zero-shot classification on ImageNet-1k, semantic segmentation on ADE20K, and both object detection and instance segmentation on MS-COCO, when pre-training a ViT-B on CC3M. We also show that Harmony outperforms SILC on detection, linear and fine-tuning classification, and outperforms other self-supervised learning methods like iBOT and MAE across all tasks evaluated. Our code is publicly available at https://github.com/MohammedSB/Harmony.

## 1 Introduction

Self-supervised and weakly-supervised pre-training have recently shown remarkable success at learning visual representations without direct supervision (Radford et al., 2021; Oquab et al., 2024; Caron et al., 2021b; Chen et al., 2020a;b; He et al., 2020; Grill et al., 2020; Zhou et al., 2022; He et al., 2021; Mu et al., 2021; Dong et al., 2023). As vision training datasets continue to scale, it becomes progressively more difficult and expensive to

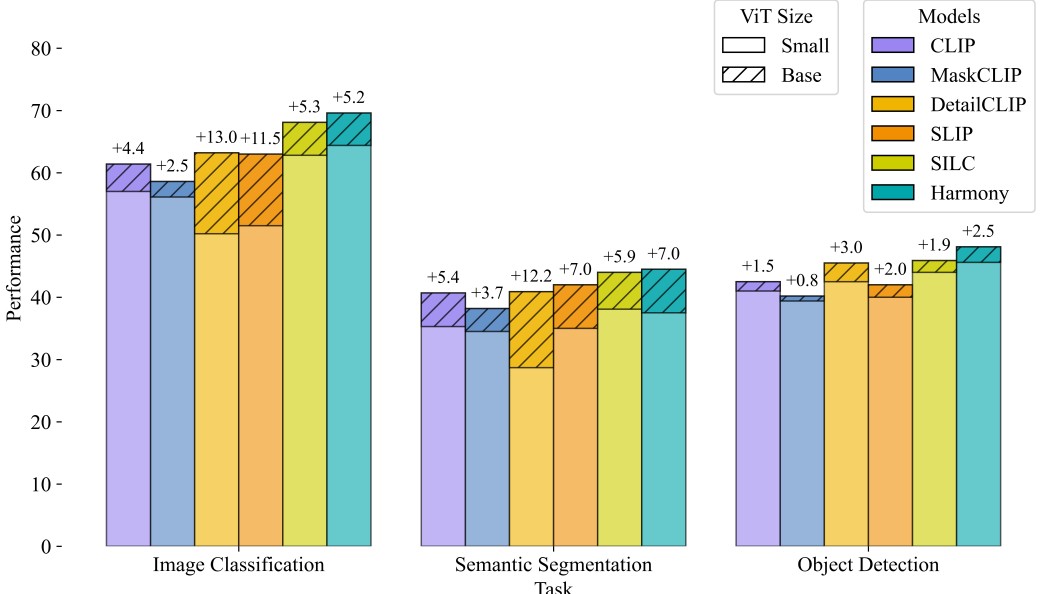

Figure 1: Performance comparison across different tasks. Harmony significantly outperforms CLIP, SLIP, MaskCLIP, and DetailCLIP across all tasks, and generally has a larger improvement in performance as we scale the ViT size, except for DetailCLIP which is low performing at ViT-S. Moreover, for classification and detection, the small version of Harmony outperforms the base version of the other methods. We used MS-COCO for object detection, ADE20K for semantic segmentation, and ImageNet for classification, using linear-probing. See Table 1 for numerical values.

provide manual supervision in the form of labels, making the development of robust self-supervised learning (SSL) and weakly-supervised learning (WSL) techniques more integral.

Weakly-supervised learning, specifically language-guided or language-supervised learning, was popularized by CLIP (Radford et al., 2021) and learns visual and textual representations using contrastive loss by maximizing the similarity between image-captions pairs and minimizing the similarity of non-paired image-captions (Radford et al., 2021; Cherti et al., 2022). Because this approach relies on semantic captions as a supervisory signal, language-supervised models are strong at high-level tasks like image classification, but significantly underperform on dense, low-level prediction tasks that require localized features (Radford et al., 2021; Wang et al., 2024; 2022). In other words, these paradigms are good at learning what objects are present in a visual input, but not where they are. One approach of introducing local information into WSL frameworks is to combine it with self-supervised learning (Dong et al., 2023; Mu et al., 2021; Yuan et al., 2021). Unlike language-supervised learning that maps across modalities (e.g. image to text), SSL maps to the same visual modality, making it more granular and localized for visual tasks (Caron et al., 2021b; He et al., 2021; Assran et al., 2023).

Recent works in self-supervised learning formulate the pre-training task as either discriminative or generative (Ozbulak et al., 2023; Doersch et al., 2016). For discriminative SSL methods, the model learns visual representations from images by differentiating between positive images pairs, and optionally repelling negative pairs (Chen et al., 2020a; Grill et al., 2020; Caron et al., 2021b; Oquab et al., 2024; He et al., 2020; Chen et al., 2020b). On the other hand, generative approaches learn visual representations by masking certain parts of an image and learning to reconstruct the missing parts, given the original image (He et al., 2021; Tong et al., 2022; Gupta et al., 2023). Because SSL methods do not require manually annotated labels, they can be used for training large neural networks on huge image datasets scraped from the web (Oquab et al., 2024).

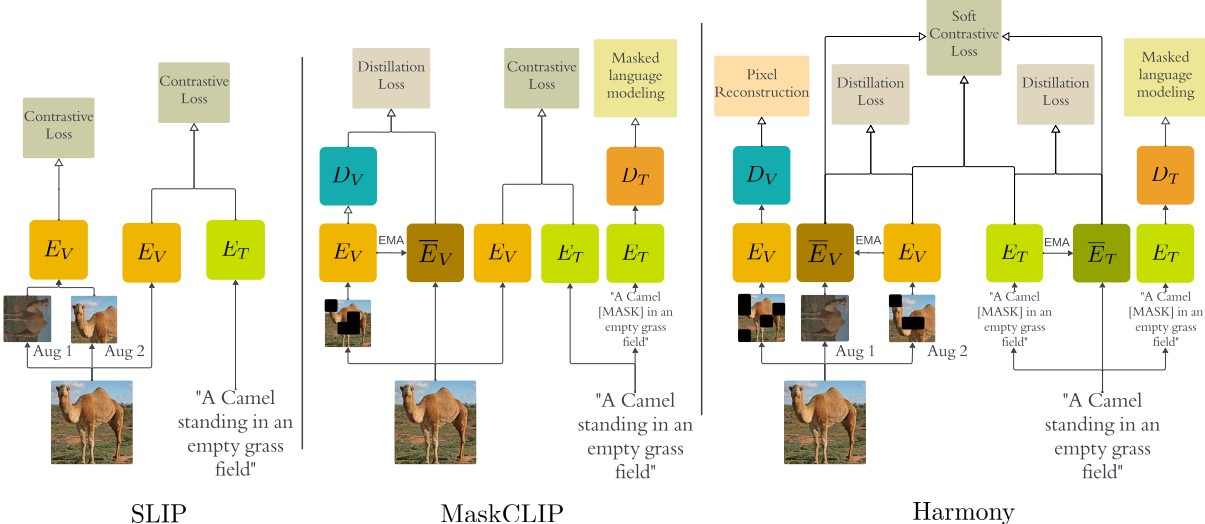

Figure 2: Harmony compared to previous methods. We show our Harmony approach next to previously leading joint methods MaskCLIP (Dong et al., 2023) and SLIP (Mu et al., 2021). Harmony optimizes five different objectives simultaneously, outperforming the other approaches across all vision downstream tasks.

Combining language-supervised and self-supervised learning has led to superior learning paradigms (Dong et al., 2023; Mu et al., 2021; Yuan et al., 2021; Naeem et al., 2024; Monsefi et al., 2024). For example, SLIP (Mu et al., 2021) combines CLIP with SimCLR, leading to better zero-shot capabilities and overall more accurate models across multiple downstream tasks. MaskCLIP (Dong et al., 2023) combines CLIP with masked self-distillation, leading to further improvements in a variety of downstream evaluations. More recently, SILC (Naeem et al., 2024) combined CLIP with DINO (Caron et al., 2021b) to learn more localized image features for dense prediction tasks. DetailCLIP (Monsefi et al., 2024) combines A-CLIP (Yang et al., 2023) with MAE (He et al., 2021) and iBOT (Zhou et al., 2022), outperforming all three methods in isolation.

Building on the aforementioned works, we present a novel framework, which we will call Harmony, that combines discriminative and generative self-supervision with language-supervised learning in order to learn general purpose visual representations from image-captions pairs gathered from the web. Our approach is designed to learn semantic visual representations useful for high-level vision tasks and fine-grained, low-level representation for dense prediction tasks at the same time.

Previous approaches like MaskCLIP (Dong et al., 2023) and SLIP (Mu et al., 2021) rely on one-to-one correspondences and hard negative examples which fail to model the inherent semantic relationship between non-paired samples (Andonian et al., 2022). This is because for a certain image, captions of other images in the batch could still describe this image with varying degrees, especially when the batch size is large. Moreover, particularly in an uncurated data setting, the caption paired with a certain image could simply be incorrect, or the description could be only loosely associated with the image. To remedy these issues, we designed our framework to utilize one-to-many relationships by incorporating soft CLIP targets, and to not rely on negative examples by using self-distillation methods like iBOT (Zhou et al., 2022) rather than SimCLR (Chen et al., 2020a) like in SLIP (Mu et al., 2021). We argue that this makes Harmony better designed for web-scraped data like CC3M (Sharma et al., 2018).

We summarize our main contributions in this paper as follows:

- We present *Harmony*, a joint self-supervised and weakly-supervised framework that learns global and local features, generalizing across vision tasks including classification, segmentation, and detection. Harmony optimizes for five different objectives simultaneously, fully leveraging each data example, which makes our method particularly suited for data-constrained settings.

- We introduce an EMA-based method for generating soft-targets for CLIP and a lightweight text self-distillation objective for learning textual representation, improving CLIP's zero-shot capabilities.

- Harmony is built to avoid negative examples in the self-supervised learning path and reduce reliance on strict one-to-one correspondences in the text-guided contrastive path, achieving superior or competitive performance with leading methods when pre-trained on the web-scraped CC3M dataset.

- We provide a unified codebase for all methods that is composable, allowing users to combine any subset of our predefined set of methods, and with our own implementation of SILC and MaskCLIP, which were not previously open-sourced.

## 2 Related Works

**Discriminative self-supervision.** Recent discriminative SSL approaches learn visual representations by discriminating between images that are positive pairs of each other, usually defined as different augmentations of the same image, and optionally pushing away negative pairs. (Grill et al., 2020; Caron et al., 2021b; He et al., 2020; Chen et al., 2020b;a). Methods like SimCLR (Chen et al., 2020a) and MoCo (He et al., 2020; Chen et al., 2020b) use contrastive learning to maximize the embedding similarity between positive pairs while simultaneously reducing the similarity of negative pairs. These approaches ignore the inherent similarity between different images and treat the discriminative problem as a one-to-one scheme, where an image is only similar to its positive pair. Other approaches like BYOL (Grill et al., 2020) and DINO (Caron et al., 2021b) use self-distillation to only maximize agreement between positive pairs, avoiding that issue. These methods use a momentum encoder, usually refereed to as a target or teacher encoder, to generate embedding targets for an online or student encoder that is trained simultaneously with the teacher encoder. The learning task in these paradigms is to maximize the similarity between embeddings from both the teacher and student encoders given positive image pairs. Self-distillation methods regularly outperform contrastive methods, and reach accuracies that are competitive with supervised learning (Caron et al., 2021b; Zhou et al., 2022; Grill et al., 2020).

**Generative self-supervision.** Rather than discriminating between images, generative SSL methods learn visual representations by masking certain parts of an images (or certain images in a video) and then learn to reconstruct the masked portions given the original signal (He et al., 2021; Tong et al., 2022; Gupta et al., 2023). This reconstruction task can be viewed as a proxy for learning about visual features from images, improving representation ability and, therefore, performance on downstream tasks. A common approach in this paradigm is called Masked Autoencoder (MAE) (He et al., 2021), which utilizes ViT's patching paradigm to mask a certain percentage of patches (usually 75%) and minimizes the L2 loss on predicted pixels from the masked patches. Generative SSL approaches are highly scalable due to their lightweight memory usage compared to discriminative approaches, and reach fine-tuning accuracies rivaling those methods (He et al., 2021).

**Joint self-supervision methods.** Many previous works have combined discriminative and generative self-supervised learning objectives, resulting in improved performance on downstream classification and semantic segmentation tasks (Chen et al., 2023; Huang et al., 2024; Mishra et al., 2022). CMAE (Huang et al., 2024) and CAE (Chen et al., 2023) combine MAE's pixel reconstruction task with contrastive learning, outperforming either method in isolation. CAN (Mishra et al., 2022) further add a noise prediction task and masks both views in the contrastive learning objective. These works highlight the fact that SSL paradigms can be complementary to each other.

**Language-guided and self-supervised learning.** Other works have combined SSL and language-guided learning to learn more generalized representations (Mu et al., 2021; Dong et al., 2023; Yuan et al., 2021). Yuan et al. (2021) combine vision-language training with contrastive SSL to learn visual representations from multimodal, image-text data. SLIP (Mu et al., 2021) builds on this work further by showing that combining SimCLR (Chen et al., 2020a) with CLIP (Radford et al., 2021) objectives outperforms both methods in isolation by relatively large margins across multiple downstream tasks. SILC (Naeem et al., 2024) replaces SimCLR with DINO in SLIP, outperforming SLIP. In addition, MaskCLIP (Dong et al., 2023) presents a masked self-distillation approach combined with contrastive vision-language pre-training,

further outperforming CLIP and other joint WSL and SSL methods. More recently, DetailCLIP (Monsefi et al., 2024) combines A-CLIP (Yang et al., 2023) with MAE and iBOT to learn more localized visual features for fine-grained tasks. We further build on these works by going an extra step of combining five different objectives, including self-distillation, pixel reconstruction, text-guided contrastive learning, and two additional text-only losses: masked-language modeling and text self-distillation.

## 3    Harmony

We introduce Harmony, a joint self-supervised and weakly-supervised framework for learning semantic and localized visual representations in the wild. The main components of our framework are shown in Figure 2. Namely we define vision student and teacher encoders, $E_V$ and $\bar{E}_V$, text student and teacher encoders, $E_T$ and $\bar{E}_T$, and vision and text decoders, $D_V$ and $D_T$. Our teacher encoders are used to generate self-distillation and soft targets, and are defined as being the exponential moving average (EMA) of the students. The vision and text decoders are used to map from embedding space to pixel and word token space, respectively. All components use the Transformer (Vaswani et al., 2023) architecture, and we use the standard vision transformer (ViT) implementation for processing images (Dosovitskiy et al., 2021).

Our framework optimizes five different objectives simultaneously with the goal of learning robust general-purpose visual representations from web-scraped images-caption pairs. Our losses are (1) text-guided contrastive learning that is identical to CLIP (Radford et al., 2021) but with added soft targets, (2) feature self-distillation following iBOT (Zhou et al., 2022), (3) pixel prediction following MAE (He et al., 2021), (4) word prediction or MLM (Devlin et al., 2019), and (5) text self-distillation, which is similar to iBOT's patch-level objective but applied on word embeddings (Zhou et al., 2022). In the following sections, we detail each objective in our framework.

### 3.1    Text-guided Contrastive Learning with Soft Targets

Our first objective in Harmony is image-text contrastive learning with soft targets. We begin by defining a vision encoder $E_V$ and a text encoder $E_T$. We attach a single layer projection head $h$ to each encoder, resulting in $h : g = h \circ f$, where $f$ is either $E_V$ or $E_T$. Given a batch of image-text pair collections $\{(v_1, t_1), (v_2, t_2), ..., (v_N, t_N)\}$ where $N$ is the batch size, we extract image and text embeddings $\mathbf{v}_i = g_v(v_i)$ and $\mathbf{t}_i = g_t(t_i)$, where $g_v$ and $g_t$ are the student vision encoder $E_V$ and student text encoder $E_T$ with the attached projection heads, respectively.

We maximize the similarity between paired embedding sets, $\mathbf{v}_i$ and $\mathbf{t}_j$ where $i = j$ and minimize the similarity between unpaired sets where $i \neq j$. More formally, we define the InfoNCE loss as our training objective van den Oord et al. (2019) following CLIP Radford et al. (2021), where $\mathcal{L}_{\text{InfoNCE}} = \mathcal{L}_v + \mathcal{L}_t$ and

$$\mathcal{L}_v = -\frac{1}{N} \sum_{i=1}^{N} \sum_{j=1}^{N} \mathrm{I}_{ij} \log P_v(\mathbf{v_i}, \mathbf{t_j}; \tau) \,. \tag{1}$$

$\mathrm{I}_{ij}$ is an element in the identity matrix $\mathbf{I_N}$ so it is set to one when $i = j$ or when the image-text embeddings are paired, and to zero otherwise. $P_v$ is the softmax function applied per image:

$$P_v(v_i, t_j; \tau) = \frac{\exp(\cos(\mathbf{v_i}, \mathbf{t_j})/\tau)}{\sum_{k=1}^{N} \exp(\cos(\mathbf{v_i}, \mathbf{t_k})/\tau)} \,. \tag{2}$$

The function $\text{sim}(\mathbf{v_i}, \mathbf{t_j})$ is the cosine similarity, $\text{sim}(\mathbf{v_i}, \mathbf{t_j}) = \mathbf{v_i}^T \mathbf{t_j}$, and $\tau$ is a learnable temperature parameter. The loss $\mathcal{L}_t$ and function $P_t$ are defined in a symmetrical way van den Oord et al. (2019). Since the above $\mathcal{L}_{\text{InfoNCE}}$ uses hard targets (1s and 0s), we will refer to it as $\mathcal{L}_{\text{Hard}}$.

Notice that $\mathcal{L}_v$ (and $\mathcal{L}_t$ given the symmetry) is the cross-entropy function $\mathcal{H}(a, b) = -a \log b$ applied across image caption pairs. Because of that, we can rewrite $\mathcal{L}_v$ as $\mathcal{H}(\mathbf{I_N}, P(\mathbf{V}\mathbf{T}^T; \tau))$ where $\mathbf{V}, \mathbf{T} \in \mathbb{R}^{N \times d}$ are matrices that contain a batch of image and text embeddings, with $d$ being the embedding size. $P$ is the vectorized version of $P_v$ and $P_t$ in Equation 2, where the cosine similarities, $\cos(\mathbf{v_i}, \mathbf{t_j})$, are calcualted through

matrix multiplication. Therefore, $\mathcal{L}_{\text{Hard}}$ can be rewritten as

$$\mathcal{L}_{\text{Hard}} = \mathcal{H}(\mathbf{I_N}, P(\mathbf{V}\mathbf{T}^T; \tau)) + \mathcal{H}(\mathbf{I_N}, P(\mathbf{T}\mathbf{V}^T; \tau)). \tag{3}$$

**Soft targets.** The loss function in the original CLIP implementation Radford et al. (2021) assumes that there is a one-to-one correspondence between image and caption pairs. This is because the target in the cross entropy function in Equation 3 is the identity matrix $\mathbf{I_N}$. This means that, given an image embedding $\mathbf{v}_i$ and a set of textual embeddings $\{\mathbf{t}_1, \mathbf{t}_2, \ldots, \mathbf{t}_N\}$, minimizing the objective can be viewed as an $N$-way multi-class classification problem. This assumption does not always accurately represent the relationship between image-captions sets, especially in the case of uncurated data. Certain captions can describe many different images with varying degrees, regardless of their original pairing, and the target should reflect some function of how semantically similar an image-caption set is.

Recent works have tried to mitigate this issue by incorporating soft instead of hard targets (Andonian et al., 2022; Gao et al., 2023; Scotti et al., 2023). This is a non-trivial task since generating soft targets assumes some pre-existing knowledge about how semantically similar image-caption pairs are. Gao et al. (2023) relies on a pre-trained object detection model to generate soft similarity targets, while Andonian et al. (2022) used a self-distillation approach to dynamically generate soft targets without pre-training. In the latter approach the same model is used to both generate soft targets and make predictions for calculating the loss, by splitting the mini-batch into student and teacher targets. Instead, we use an EMA self-distillation method for generating soft-targets, motivated by the fact that our framework already defines an EMA model for feature self-distillation in Section 3.2, so the addition of soft targets comes at little computational expense.

To generate soft targets, we define vision and text teacher models, $\bar{E}_V$ and $\bar{E}_T$, respectively, and their corresponding $\bar{g}_v$ and $\bar{g}_t$, which are the encoders with the attached projection heads. We do not propagate gradients through the parameters of either $\bar{E}_V$ or $\bar{E}_T$ and instead update their weights using the EMA of the students. Given the same image-text pair collections $\{(v_1, t_1), (v_2, t_2), \ldots, (v_N, t_N)\}$, we generate embedding targets $\bar{\mathbf{v}}_i = \bar{g}_v(v_i)$ and $\bar{\mathbf{t}}_i = \bar{g}_t(t_i)$. We can represent these targets as the matrices $\bar{\mathbf{V}}$ and $\bar{\mathbf{T}}$ like in Equation 3, with $\mathbf{A}_V = P(\bar{\mathbf{V}}\bar{\mathbf{T}}^T; \bar{\tau})$ and $\mathbf{A}_T = P(\bar{\mathbf{T}}\bar{\mathbf{V}}^T; \bar{\tau})$. Our soft CLIP loss, $\mathcal{L}_{\text{Soft}}$ is then defined as

$$\mathcal{L}_{\text{Soft}} = \mathcal{H}(\mathbf{A}_V, P(\mathbf{V}\mathbf{T}^T; \tau)) + \mathcal{H}(\mathbf{A}_T, P(\mathbf{T}\mathbf{V}^T; \tau)). \tag{4}$$

We set the teacher temperature $\bar{\tau}$ to 0.1. Our final contrastive loss is a the sum of the two losses $\mathcal{L}_{\text{Hard}}$ and $\mathcal{L}_{\text{Soft}}$, where we progressively increase the influence of the soft loss $\mathcal{L}_{\text{Soft}}$ throughout training. In other words, the contrastive loss is defined as

$$\mathcal{L}_C = \alpha_c \mathcal{L}_{\text{Hard}} + (1 - \alpha_c)\mathcal{L}_{\text{Soft}}. \tag{5}$$

We start with $\alpha_c = 1$ and progressively decrease it to $\alpha_c = 0.2$ using a cosine scheduler in the first 10 epochs of pre-training.

## 3.2 Feature Self-distillation

On top of the contrastive objective $\mathcal{L}_C$, we add a self-supervised feature self-distillation loss following iBOT (Zhou et al., 2022), which we find can significantly boost performance across all downstream tasks evaluated (see Table 4). This loss consists of both global and local objectives that go hand-in-hand to learn generalized visual features. Conceptually, the goal of this loss in our framework is to learn visual features that might not be described in the caption of the contrastive loss, and learn more localized features by adding a local objective that operates at the patch level.

**Global objective.** Following (Caron et al., 2021b; Oquab et al., 2024; Grill et al., 2020; Zhou et al., 2022) we utilize a teacher encoder $\bar{E}_V$ that is defined as the EMA of a student encoder $E_V$, which is trained with gradient optimization. The teacher and student models, $\bar{E}_V$ and $E_V$, are the same models used in the contrastive loss from Section 3.1. Just like in (Caron et al., 2021b; Zhou et al., 2022), we attach a multi-layer

perceptron (MLP) projection head $h$, such that $h : g = h \circ f$, where $f$ is either $\bar{E}_V$ or $E_V$, resulting in $\bar{g}$ and $g$ models, respectively. Given an input image $x$ we generate two different augmentations, $x_1$ and $x_2$, defined as random crops or views of $x$. We feed these two augmentations to $\bar{g}$ and $g$, and optimize the following loss function:

$$\mathcal{L}_{\text{CLS}} = -\frac{\mathcal{H}(\bar{P}(x_1), P(x_2)) + \mathcal{H}(\bar{P}(x_2), P(x_1))}{2}, \tag{6}$$

where $\mathcal{H}$ is the cross entropy function and $P$ is a softmax function, defined as

$$P(x) = \frac{\exp(g(x)/\tau)}{\sum_{k=1}^{K} \exp(g(x_k)/\tau)}. \tag{7}$$

$\tau$ is a non-trainable temperature parameter that controls output distribution sharpness. $\bar{P}(x)$ is defined in the same way but with $\bar{g}$ instead of $g$ and $\bar{\tau}$ instead of $\tau$.

**Local objective.** Equation 6 is applied on the class token (CLS) of the vision transformer (Dosovitskiy et al., 2021), which is why we refer to it as $\mathcal{L}_{\text{CLS}}$. Since CLS tokens aggregate information across different patches into a single token, minimizing $\mathcal{L}_{\text{CLS}}$ learns more global features, sometimes disregarding granular details that are helpful for dense prediction tasks like semantic segmentation. On top of this global-level objective we aim to learn localized features by employing a patch-level loss using masked image modeling (MIM), following iBOT (Zhou et al., 2022). More precisely, given the sequence of image tokens $x = \{x_1, x_2, \ldots, x_N\}$ being processed by a ViT, we sample from the masking set $m \in \{0, 1\}^N$ according to a ratio $r$. If $m_i = 1$, we replace the original $x_i$ with a special mask token $x_m$, resulting in a masked view $\hat{x}$ formalized as $\hat{x} \triangleq \{\hat{x}_i ((1 - m_i)x_i + m_i x_m)\}_{i=1}^{N}$ (Zhou et al., 2022). We subsequently feed $x$ and $\hat{x}$ to $\bar{g}$ and $g$, respectively, optimizing the loss function over all patch tokens:

$$\mathcal{L}_{\text{MIM}} = -\sum_{i=1}^{N} m_i \, \mathcal{H}(\bar{P}(x_i), P(\hat{x}_i)). \tag{8}$$

As shown in (Zhou et al., 2022), adding the patch-level loss $\mathcal{L}_{\text{MIM}}$ on top of $\mathcal{L}_{\text{CLS}}$, which is originally from DINO (Caron et al., 2021b), improves downstream performance on both dense prediction and classification tasks.

Moreover, we also minimize the cross entropy loss between the embeddings of $x_1$ and $x_2$ extracted by the teacher model $\bar{E}_V$ and embeddings of smaller, local crops $\{y_1, y_2, \ldots, y_L\}$ extracted by the student model $E_V$ and generated using a multi-crop augmentation strategy (Caron et al., 2021a), which is shown to increase performance in (Caron et al., 2021b; Zhou et al., 2022). This multi-crop optimization is added only to $\mathcal{L}_{\text{CLS}}$.

The final self-distillation loss $\mathcal{L}_D$ is as the average of $\mathcal{L}_{\text{CLS}}$ and $\mathcal{L}_{\text{MIM}}$, i.e.,

$$\mathcal{L}_D = \frac{1}{2}(\mathcal{L}_{\text{CLS}} + \mathcal{L}_{\text{MIM}}). \tag{9}$$

### 3.3 Pixel Reconstruction

The $\mathcal{L}_D$ loss is applied on the feature space. On top of this, we add another pixel level loss for learning more granular features, which we find slightly improve performance for segmentation tasks (see Table 4). Specifically, we follow the approach in MAE (He et al., 2021) in that, given a sequence of patch token $\{x_1, x_2, \ldots, x_N\}$, we mask (remove) $P$ number of the patch tokens at random (as opposed to replacing them with $x_m$ like in Section 3.2). We feed the remaining $(L - P) + 1$ of tokens, where the 1 is the $x_{\text{CLS}}$ token and $L$ is the total number of patches in the original images, to $E_V$, which is the same encoder used in the contrastive objective (Section 3.1) and feature self-distillation (Section 3.2). We end up with a sequence of image embeddings $e = \{e_{\text{CLS}}, e_1, e_2, \ldots, e_M\}$, where $M = L - P + 1$. We then add $P$ number of mask tokens $x_m$ to $e$ in the same position (or index) they were removed from to obtain the masked embedding $\hat{e}$. Subsequently, $\hat{e}$ is passed to a decoder $D_V$, which will up-sample the embeddings from $d_D$ to the $H \times W$, where $d_D$ is the embedding size of $D_V$, $H$ and $W$ are the height and width of the original patch size, respectively. We set $P = L \times 0.75$, following He et al. (2021).

The loss is then calculated as the Mean Squared Error between the predicted pixels $p_i$ and the L2 normalized pixels of the original target in a patch $\hat{p}_i$. In other words, the loss becomes:

$$\mathcal{L}_R = \frac{1}{L} \sum_{i=1}^{L} \text{m}_\text{i} \, (p_i - \hat{p}_i)^2 \,. \tag{10}$$

### 3.4 MLM and Text Self-distillation

Even though the goal of this work is visual representation learning, improving textual representations can have a significant impact on zero-shot evaluations as supported by results from Dong et al. (2023) and our Table 4. As a result, we added two textual objectives that the text encoder $E_T$ will now optimize on top of the contrastive loss. These objectives are masked language modeling and text self-distillation.

**Masked language modeling.** The first additional textual objective is masked language modeling, as described in BERT (Devlin et al., 2019), which predicts masked words given the context of surrounding words. MLM can be viewed analogously as pixel prediction, in the sense that both optimize outputs that are in the same level as original inputs (tokenized words in the context of MLM and pixels in pixel prediction).

We feed a masked view $\hat{t}$ of a tokenized caption $t$ to the student text encoder $E_T$, where word tokens are randomly masked using a Bernoulli distribution $m \sim \text{Bernoulli}(p)$ with $p = 0.2$, with $m$ being the mask vector that is applied on a caption $t$ to generate $\hat{t}$. Unlike BERT (Devlin et al., 2019), we don't replace words with random words, nor keep some predicted words unchanged. Masked word tokens are replaced with a mask token $t_\text{m}$ to generate $\hat{t}$. Our final MLM loss is then formulated as

$$\mathcal{L}_M = \sum_{i=1}^{C} \text{m}_\text{i} \, \mathcal{H}(t_i, D_T(E_T(\hat{t}_i))) \,, \tag{11}$$

where $C$ is the context length of the transformer model $E_V$ and $\text{m}_\text{i} = 1$ only if $\hat{t}_i$ is masked.

**Text self-distillation.** On top of the MLM loss, we add a self-distillation loss by utilizing the student and teacher text encoders, $E_T$ and $\bar{E}_T$, motivated by iBOT's patch-level objective (Zhou et al., 2022). Unlike the MLM loss, this objective will function at the embedding level, which offers softer targets for the student text encoder compared to MLM. We attach MLP projection heads to $E_T$ and $\bar{E}_T$, resulting in $g$ and $\bar{g}$, respectively. We then optimize the equation

$$\mathcal{L}_{TD} = -\sum_{i=1}^{C} \text{m}_\text{i} \, \mathcal{H}(\bar{P}(t_i), P(\hat{t}_i)) \,, \tag{12}$$

where $P$ and $\bar{P}$ are the softmax functions in Equation 7, but with the text instead of vision encoders.

### 3.5 Harmony's Objective

Our finalized framework, Harmony, optimizes the five described objectives simultaneously. In other words, our final loss is a linear combination of all five losses or the equation:

$$\mathcal{L}_H = \mathcal{L}_C + \alpha\mathcal{L}_D + \beta\mathcal{L}_R + \gamma\mathcal{L}_M + \delta\mathcal{L}_{TD} \,. \tag{13}$$

The parameters $\alpha$, $\beta$, $\gamma$, and $\delta$ allow for weighting of the different losses. However, in our experiments (see Table 6), choosing identical weights has usually been sufficient.

## 4 Experiments

Here, we experimentally evaluate Harmony against the baseline methods CLIP (Johnson et al., 2016), SigLIP (Zhai et al., 2023), iBOT (Zhou et al., 2022), and MAE (He et al., 2021), as well as previously leading joint SSL and WSL methods, SLIP (Mu et al., 2021), SILC (Naeem et al., 2024), MaskCLIP (Dong et al., 2023), and DetailCLIP (Monsefi et al., 2024). We start in Section 4.1 by describing our model architecture and training setup, then we present our results in Section 4.2, and finally end with an ablation and hyper-parameter tuning study in Section 4.3.

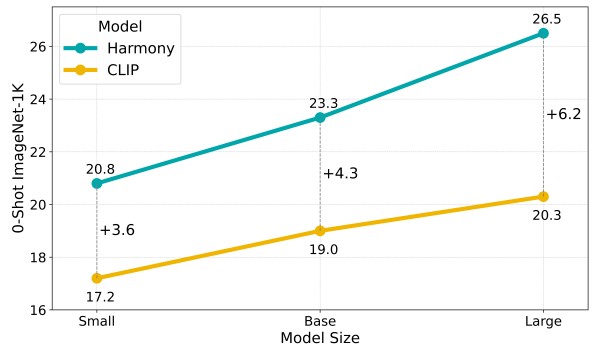 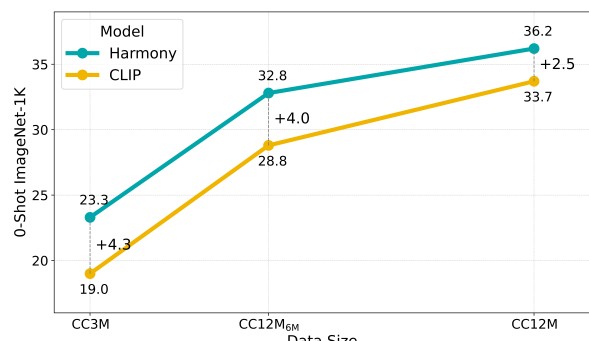

(a) Scaling ViT size. The gap in 0-shot performance between Harmony and CLIP increases as we increase model size, suggesting that Harmony scales better with respect to model size.

(b) Scaling data size. The performance gap between Harmony and CLIP narrows as dataset size increases. CC12M$_{6M}$ indicates using 6M images from CC12M. ViT-B was used for all experiments.

Figure 3: Zero-shot performance comparison between Harmony and CLIP when scaling ViT size and data. (a) With increasing model size, Harmony's advantage grows. (b) With increasing dataset size, the performance gap narrows, though Harmony maintains an edge. Appendix B shows zero-shot results on CC12M across 8 other datasets.

## 4.1 Training Setup

**Model architecture.** Six transformer networks (Vaswani et al., 2023), three of which are vision transformers (Dosovitskiy et al., 2021), of different sizes and configurations, make up our final framework: (1, 2) student and teacher vision encoders, $E_V$ and $\bar{E}_V$; (3, 4) student and teacher text encoders, $E_T$ and $\bar{E}_T$; (5) vision decoder $D_V$; and (6) text decoder $D_T$. For the vision encoders, $E_V$ and $\bar{E}_V$, we use ViT sizes small (S), base (B), and large (L). For the text encoders, $E_T$ and $\bar{E}_T$, we use 12 layers, 512 embedding dimensions, and 8 heads, following CLIP (Radford et al., 2021). The number of text tokens is fixed to 77 with necessary truncations or paddings. Moreover, following He et al. (2021), we define our vision decoder $D_V$ as an 8-layer, 512-embedding dimension, and 16-head ViT. Finally, our text decoder $D_T$, is made of up 4 layers, 512 embedding dimension, and 8 heads.

**Pre-training.** We pre-train a ViT-S, ViT-B, and ViT-L using Harmony on CC3M for 25 epochs. We use an AdamW optimizer (Loshchilov & Hutter, 2019) with an $\epsilon$ value of $1e^{-6}$ to improve training stability using mixed precision (Micikevicius et al., 2018). We set the number of global crops per given image to 2 and the local crops to 8. We use a masking ratio of 75% for the pixel prediction task and pre-train up to 16 32GB V100 GPUs. Further information about our pre-training setup can be found in Appendix A.

**Downstream.** To asses the generalizability of learned features using our proposed method, we evaluate our model on a variety of datasets, including ImageNet-1k (Russakovsky et al., 2015) for fine-tuning, linear-probing, and zero-shot classifications, ADE20K (Zhou et al., 2017) for semantic segmentation, MS-COCO (Lin et al., 2015) for object detection and instance segmentation, as well as many other datasets for zero-shot evaluations (see Table 2). We fine-tune for 100 epochs on ImageNet-1k and for 160k iterations for ADE20K, using UperNet (Xiao et al., 2018). For object detection and instance segmentation, we use a Cascade Mask R-CNN (Cai & Vasconcelos, 2019) and fine-tune for 12 epochs. Further details on downstream settings can be found in Appendix A.

## 4.2 Results

We begin our evaluation of Harmony by analyzing its downstream performance for a variety of visual tasks in different settings. We start in Table 1 by presenting our main result for classification, segmentation, and detection tasks. We show that Harmony significantly outperforms CLIP and SigLIP and outperforms iBOT (Zhou et al., 2022) and MAE (He et al., 2021) in all high-level and low-level tasks evaluated. We also show that Harmony outperforms SLIP (Mu et al., 2021), MaskCLIP (Dong et al., 2023), and DetailCLIP

Table 1: Comparing Harmony against baseline SSL and previous best methods. All methods are pre-trained on CC3M for 25 epochs. We used the same pre-training settings for MAE, iBOT, CLIP, SigLIP, SILC and Harmony. To reproduce SLIP, and DetailCLIP, we used their public code base (Mu et al., 2021; Monsefi et al., 2024). We reproduced MaskCLIP and SLIC ourselves since their code was not publicly available (see Appendix C for further detail on MaskCLIP). FT: fine-tuning; LIN: linear-probing; SSEG: semantic segmentation; DET: object detection; ISEG: instance segmentation.

| Method | ViT | INet-1K | | | ADE20K | COCO | |
|---|---|---|---|---|---|---|---|
| | | 0-Shot | LIN | FT | SSEG | DET | ISEG |
| MAE | S | — | 14.6 | 70.9 | 26.9 | 28.9 | 25.6 |
| iBOT | S | — | 58.2 | 79.5 | 37.0 | 44.9 | 39.0 |
| CLIP | S | 17.2 | 57.0 | 76.7 | 35.3 | 41.0 | 35.7 |
| SigLIP | S | 16.0 | 57.0 | 76.8 | 36.2 | 40.8 | 35.6 |
| MaskCLIP | S | 17.7 | 56.1 | 77.0 | 34.5 | 39.4 | 34.5 |
| DetailCLIP | S | 18.2 | 50.2 | 77.7 | 28.7 | 42.5 | 37.1 |
| SLIP | S | 18.1 | 51.5 | 76.5 | 35.0 | 40.0 | 35.0 |
| SILC | S | 20.1 | 62.8 | 78.2 | 38.1 | 44.0 | 38.3 |
| CLIP | B | 19.0 | 61.4 | 79.8 | 40.7 | 42.5 | 37.0 |
| SigLIP | B | 18.0 | 61.8 | 79.9 | 40.7 | 42.5 | 37.0 |
| MaskCLIP | B | 19.1 | 58.6 | 79.5 | 38.2 | 40.2 | 35.1 |
| DetailCLIP | B | 21.1 | 63.2 | 81.6 | 40.9 | 45.5 | 39.5 |
| SLIP | B | 19.1 | 63.0 | 80.3 | 42.0 | 42.0 | 36.8 |
| SILC | B | **23.3** | 68.1 | 81.6 | 44.0 | 45.9 | 39.8 |
| Harmony | S | 20.8 | 64.4 | 79.5 | 37.5 | 45.6 | 39.5 |
| | B | **23.3** | **69.6** | **82.4** | **44.5** | **48.1** | **41.6** |

Table 2: Zero-shot evaluations. We compare ViT-B Harmony against CLIP, SLIP, SILC, MaskCLIP, and DetailCLIP on general and natural image benchmarks for 0-shot classification. Harmony outperforms the other methods on average.

| | Average | Cal101[1] | STL-10[2] | Kin700[3] | MNIST[4] | CLEVR[5] | INet-A[6] | INet-O[7] | INet-R[8] |
|---|---|---|---|---|---|---|---|---|---|
| CLIP | 28.9 | 53.9 | 87.2 | 17.4 | 10.1 | 12.8 | 4.5 | 24.2 | 21.4 |
| MaskCLIP | 28.2 | 49.8 | 84.9 | 15.7 | 12.1 | 14.2 | 4.0 | 24.8 | 20.0 |
| DetailCLIP | 29.9 | 55.2 | 89.1 | 17.1 | 11.6 | 11.6 | 4.4 | 27.3 | 22.9 |
| SLIP | 28.3 | 50.5 | 85.2 | 16.2 | 10.8 | 13.3 | 4.9 | 23.7 | 22.2 |
| SILC | 32.7 | 58.6 | 91.5 | 19.8 | 15.1 | 12.4 | 6.9 | 29.2 | 28.4 |
| Harmony | 32.9 | 60.3 | 94.3 | 22.5 | 10.1 | 12.3 | 7.6 | 26.1 | 29.8 |

(Monsefi et al., 2024), which are joint methods similar to ours. We also show that Harmony is compettive with SILC (Naeem et al., 2024) on zero-shot classification, but outperforms SILC on other tasks such detection, linear and fine-tuning classification. Interestingly, MAE performs substantially worse than the other methods, which could be due to the distribution of the web-scraped data in CC3M. Even then, using the pixel reconstruction objective in MAE still complements the features for Harmony (shown as an ablation in Table 4). Moreover, SLIP performs worse than expected on classification tasks, which could be due to the small batch size for SimCLR's contrastive loss (Chen et al., 2020a) (768 vs 4096 in the original SLIP (Mu et al., 2021)). This issue is less prominent in methods that use self-distillation such as Harmony and SILC (Dong et al., 2023; Naeem et al., 2024) because they do not rely on negative examples in the batch in their self-supervised losses.

**Scaling Harmony.** In Figure 3 we present the performance of Harmony and CLIP as we scale model sizes from small to base and large, and as we scale data size from CC3M to CC12M. As we scale the model, the performance gap increases at an increasing rate, which is a sign that Harmony scales better than CLIP with respect to ViT size. The same trend is also shown in Figure 1, going from ViT-S to ViT-B for other tasks.

When scaling the dataset size, the performance gap between Harmony and CLIP narrows. This reduction is expected, as improvements become harder to achieve at higher accuracy levels. However, it may also suggest that CLIP benefits more from larger datasets and begins to close the gap with Harmony as the data scale increases. Appendix B shows zero-shot results on CC12M across 8 other datasets.

**Zero-shot evaluations across datasets.** Moreover, we compare the 0-shot capabilities of Harmony to CLIP, SLIP, SILC, MaskCLIP, and DetailCLIP across diverse datasets. We focus on general and natural datasets (as opposed to domain specific like StanfordCars (Yang et al., 2015) and FGVC-Aircraft (Maji et al., 2013)), and ImageNet robustness datasets like ImageNet-R. We present our results in Table 2. Harmony outperforms all other methods in 5 out of the 8 datasets, and performs better on average compared to all methods except SILC, where it is competitive. We show qualitative examples between Harmony's and CLIP's zero-shot predictions in Appendix 4.

**Retrieval.** We also evaluate Harmony for image-text zero-shot retrievals on MS-COCO (Lin et al., 2015) and Flickr30K (Young et al., 2014). The results are presented in Table 3. We observe that for top 5 (R@5) and top 10 retrievals, the gap between Harmony and the other methods grows rapidly, indicating that our method is more robust, since it still generates higher probabilities for the correct retrieval, even if it can't retrieve it within the first examples.

---

[1]Li et al. (2022)
[2]Coates et al. (2011)
[3]Carreira et al. (2022)
[4]Lecun et al. (1998)
[5]Johnson et al. (2016)
[6]Hendrycks et al. (2021b)
[7]Hendrycks et al. (2021b)
[8]Hendrycks et al. (2021a)

Table 3: Results of zero-shot image-text retrieval on Flickr30K and MS-COCO datasets.

| | Flickr30K | | | | | | MS-COCO | | | | | |
| | Image-to-Text | | | Text-to-Image | | | Image-to-Text | | | Text-to-Image | | |
| | R@1 | R@5 | R@10 | R@1 | R@5 | R@10 | R@1 | R@5 | R@10 | R@1 | R@5 | R@10 |
|---|---|---|---|---|---|---|---|---|---|---|---|---|
| CLIP | 4.9 | 13.8 | 20.2 | 4.6 | 12.6 | 18.1 | 15.4 | 36.2 | 48.0 | 13.8 | 32.9 | 43.9 |
| MaskCLIP | 4.8 | 14.0 | 20.5 | 4.7 | 12.5 | 17.9 | 16.1 | 36.3 | 48.1 | 13.4 | 32.0 | 42.8 |
| DetailCLIP | 6.4 | 16.8 | 24.2 | 5.8 | 14.8 | 20.7 | 17.9 | 39.4 | 51.3 | 14.8 | 33.7 | 44.8 |
| SLIP | 5.4 | 15.4 | 22.3 | 5.0 | 13.5 | 19.1 | 15.4 | 38.0 | 50.3 | 13.4 | 32.3 | 43.2 |
| SILC | 7.6 | 20.4 | 28.9 | 7.3 | 18.4 | 25.4 | 20.6 | 44.5 | 56.5 | 17.9 | 39.7 | 51.4 |
| Harmony | 9.8 | 23.7 | 32.6 | 8.6 | 20.7 | 28.1 | 22.1 | 46.9 | 58.7 | 19.1 | 42.1 | 53.8 |

Table 4: Ablation study for Harmony. The upper section represents the addition of the main three objectives of our framework. The feature prediction task ($\mathcal{L}_D$) is described in Section 3.2, soft targets ($\mathcal{L}_{\text{Soft}}$) in Section 3.1 and pixel reconstruction ($\mathcal{L}_R$) in Section 3.3. The lower part shows the addition of the two text losses: Masked language modeling ($\mathcal{L}_M$) and text self-distillation ($\mathcal{L}_{TD}$). We train a ViT-S for 25 epochs and a batch size 768 for all ablations. [†]Text losses are compared to the original CLIP baseline.

| Method | Compute | | | INet-1K | | | ADE20K |
| | Mem | Time | 0-Shot | FT | LIN | SSEG |
|---|---|---|---|---|---|---|
| CLIP | 1.0x | 1.0x | 17.2 | 76.7 | 57.0 | 36.5 |
| $+ \mathcal{L}_D$ | 2.3x | 3.0x | 19.8 $\uparrow 2.6$ | 78.2 $\uparrow 1.5$ | 61.7 $\uparrow 4.7$ | 37.9 $\uparrow 1.4$ |
| $+ \mathcal{L}_{\text{Soft}}$ | 2.3x | 3.2x | 20.4 $\uparrow 0.6$ | 79.4 $\uparrow 1.2$ | 64.3 $\uparrow 2.6$ | 37.0 $\downarrow 0.9$ |
| $+ \mathcal{L}_R$ | 2.4x | 3.9x | 20.4 $\uparrow 0.0$ | 79.5 $\uparrow 0.1$ | 64.4 $\uparrow 0.1$ | 38.0 $\uparrow 1.0$ |
| Text [†] | | | | | | |
| $+ \mathcal{L}_M$ | 1.1x | 1.3x | 17.7 $\uparrow 0.5$ | 76.8 $\uparrow 0.1$ | — | 35.8 $\downarrow 0.7$ |
| $+ \mathcal{L}_{TD}$ | 1.1x | 1.4x | 18.6 $\uparrow 0.9$ | 77.1 $\uparrow 0.3$ | — | 36.4 $\uparrow 0.6$ |
| $= \mathcal{L}_H$ | 2.5x | 4.2x | 20.8 $\uparrow 3.6$ | 79.5 $\uparrow 2.8$ | 64.4 $\uparrow 7.4$ | 37.5 $\uparrow 1.0$ |

Table 5: Equal compute comparison. Performance of CLIP, SLIP, and MaskCLIP when trained with the same hyperparameters and the same computational resources (light gray) as Harmony. We saved a checkpoint every 10 epochs and reported results for the highest performance on zero-shot classification, because the models were overfitting due to the larger number of epochs. Total memory is calculated across all GPUs used, and GPU hour is calculated for a single GPU.

| Method | GPU Hour | Total Memory (GB) | Batch Size | Epochs | 0-Shot | FT |
|---|---|---|---|---|---|---|
| CLIP | 90 | 76 | 768 | 25 | 17.2 | 76.7 |
| CLIP | 413 | 209 | 2400 | 120 | 17.9 | 77.7 |
| SLIP | 167 | 184 | 768 | 25 | 18.1 | 76.5 |
| SLIP | 426 | 240 | 1024 | 65 | 18.4 | 77.7 |
| MaskCLIP | 105 | 197 | 768 | 25 | 17.0 | 77.0 |
| MaskCLIP | 417 | 202 | 800 | 100 | 19.6 | 78.4 |
| Harmony | 414 | 219 | 768 | 25 | 20.8 | 79.5 |

## 4.3 Ablations

**Building Harmony.** To quantify the improvements from each objective in Harmony, we rebuild Harmony's architecture step-by-step, starting at CLIP (Radford et al., 2021). The results for pre-training a CLIP on CC3M (Sharma et al., 2018) are shown in Table 4. We continue by adding a visual feature prediction task in the form of self-distillation ($\mathcal{L}_D$). This substantially boosts the accuracies across all tasks. Subsequently, we add soft targets ($\mathcal{L}_{\text{Soft}}$) by utilizing the same teacher network $\bar{E}_V$ used in the feature prediction, further increasing performance. However, we notice a degradation in the SSEG performance. To remedy this, we add the pixel reconstruction loss ($\mathcal{L}_R$) from MAE (He et al., 2021), which operates in the pixel space, leading to more granular pixel-level features being learned, increasing SSEG closer its value after feature prediction.

Moreover, we investigate the effect of adding our two text losses to the original CLIP. Starting from the same baseline show in Table 4, adding both MLM ($\mathcal{L}_M$) and text self-distillation ($\mathcal{L}_{TD}$) objectives increases the 0-shot classification by +1.4% compared to CLIP. This is likely due to a boost in representation ability of the text encoder $E_T$, and the introduction of an EMA teacher text encoder $\bar{E}_T$. However, even in downstream tasks that do not utilize text encoders like FT, there is still a slight increase in performance, likely due to the making the contrastive learning task more meaningful by enhancing textual representations, leading to more reflective similarity matrices in Equation 3.

**Hyper-parameter tuning.** We tune the introduced $\alpha_c$ parameter of Equation 5 that controls the weight of soft and hard CLIP targets, and the parameters in Equation 13 that control the influence of each objective in Harmony. In Appendix D, we provide additional experiments such as the effect of (1) passing the two global crops compared to a single image to our pixel reconstruction objective, (2) using iBOT's data augmentation with CLIP, (3) using a mask scheduler to change the MAE mask ratio throughout training, and (4) masking the images passed to CLIP (Li et al., 2023; Yang et al., 2023).

*Influence of $\alpha_c$ for soft targets.* For the $\alpha_c$ parameter in the contrastive loss, we compare two situations: A higher alpha value ($\alpha_c = 0.2$) reached early in the training (10 epochs) results in a 0-shot accuracy of 20.5, while a lower alpha ($\alpha_c = 0.05$), reached later in the training (15 epochs) results in a 0-shot accuracy of 16.6.

*Loss weight.* In Table 6, we adjust the weight parameter for each of the four loss functions added on top of the contrastive loss. We observe slight performance changes, with setting the weight to one generally performing better.

*Batch size.* We investigated the effect of changing the batch size for Harmony, using a ViT-S. Table 7 shows the results. The performance for 0-shot classification gradually increases as we go from a batch size of 512 to 1024, but then decreases when we go to 2048.

**Equal compute.** As shown in Table 4, Harmony uses 2.5 times more compute and takes 4.2 times more time to train for the same number of epochs, compared to CLIP, SLIP and MaskCLIP. This makes a direct

Table 6: ImageNet-1k 0-shot accuracy for different Loss Weighting. We observe that setting the weight to one performs slightly better. We compare CLIP + iBOT for $\alpha$, CLIP + MAE for $\beta$, CLIP + MLM for $\gamma$, and CLIP + MLM + TextDist for $\delta$ with $\gamma$ set to 1.

| Weight | $\alpha$ | | $\beta$ | | $\gamma$ | | $\delta$ | |
|---|---|---|---|---|---|---|---|---|
| | 0.5 | 1 | 0.5 | 1 | 0.1 | 1 | 0.2 | 1 |
| 0-Shot | 19.1 | **19.2** | 16.8 | **17.2** | 17.1 | **17.3** | 17.8 | **18.0** |

comparison between the two methods, while controlling for the number of epochs is unintuitive. Instead, in Table 5, we control for memory and time rather than the number of epochs. We increase the batch size to use more memory for the other methods. Harmony still significantly outperforms the other methods, which seem to plateau after 25 epochs, increasing their accuracy only by 1% or less (except MaskCLIP). For each run, checkpoints were saved every 10 epochs, and only the best result is shown, because the models were overfitting due to the larger number of epochs.

Table 7: Influence of changing Harmony's batch size.

| Batch Size | 512 | 768 | 1024 | 2048 |
|---|---|---|---|---|
| 0-Shot | 20.3 | 20.8 | 21.1 | 20.6 |

## 5 Conclusion

We present Harmony, a joint self-supervised and weakly-supervised method for learning generalized visual features from web-scraped data, introducing a soft loss and a text self-distillation method. Harmony outperforms or is competitive with leading methods and baselines across classification, segmentation, and detection tasks, highlighting how our multiple training objectives can complement each other to learn stronger visual representations.

## 6 Acknowledgement

We thank Paul Scotti from Stability AI for the discussion on soft targets for the contrastive loss. We also thank Yifan Yang and Weiquan Huan for sharing their retrieval evaluation code and for general discussion about attentive masking. The KAUST Supercomputing Lab has been gratefully acknowledged.

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

# A Experimental Details

**Pre-training.**

We use the AdamW optimizer (Loshchilov & Hutter, 2019), and mixed precision with FP16, in all our pre-training experiments. Following DINO (Caron et al., 2021b) and iBOT (Zhou et al., 2022) we scale the learning rate with the batch size through $lr = 5e^{-4} \times batchsize/256$. The learning rate is ramped up linearly from 0 to the value in the formula in the first 3 epochs, then decreases to $1e^{-6}$ using a cosine scheduler, by the end of training. For iBOT and DINO heads, we set the output dimension to 8192, and we share weights in both the iBOT and DINO heads for both the student and teacher. We also normalize the last layer for stability, and set the momentum parameter for the teacher to 0.996. For the masked image modeling in iBOT (Zhou et al., 2022), we follow the settings in their ViT-B ImageNet-22k experiments. Specifically, the prediction ratio is selected randomly as either 0 and 0.3 plus an added variation of $\pm 0.2$. Masking is performed block-wise. For the CLS objective in Equation 6, we set the temperature parameter $\tau$ to 0.04. For the MLM objective shown in Equation 8, the temperature parameter is linearly scaled from 0.04 to 0.07 in 10 epochs. We use a weight decay that is scaled with a cosine scheduler from 0.04 to 0.4, and use gradient clipping with a max gradient norm value of 3.0.

For data augmentation, we likewise follow the settings in iBOT (Zhou et al., 2022). For the global crops, we use a cropping with a scale uniformly sampled from $[0.32, 1]$. In all our experiments, we use 8 local crops in the global objective, with cropping scale sampled from $[0.05, 0.32]$. For the first global crop, we always apply a Gaussian blur, while we randomly apply it to the second crop with a probability of 0.1. We also randomly solarize the second crop with a probability of 0.2 and threshold of 128. We also apply a horizontal flipping to both crops with a probability of 0.5, a color jitter with probability of 0.8, and grayscaling with probability 0.2. For the local crops, we take random crops of size $96 \times 96$, and apply the same horizontal flipping and color jitter. We also apply a Gaussian blur with probability 0.5.

For the MAE pixel reconstruction objective, we randomly mask (remove) 75% of all patch tokens following He et al. (2021). We reconstruct both global crops from iBOT (see Table 12). The final pixel reconstruction loss is then calculated as the addition of the loss of both global crops. For the MLM, objective, each token is randomly masked with 20% probability.

For all ViT-S and ViT-B experiments, we use between 8 and 16 V100 GPUS running for a day. For the ViT-L run, we use 16 80GB A100 GPUs instead, running for a day.

**Fine-tuning on ImageNet-1k.** For fine-tuning evaluations, we use the pipeline in iBOT (Zhou et al., 2022), which follows the fine-tuning protocol from BEiT (Bao et al., 2022). We fine-tune for 100 epochs in all experiments using the AdamW (Loshchilov & Hutter, 2019) optimizer, with a batch size of 1024. We linearly warmup the learning rate to $4e^{-3}$ in 20 epochs then lower it with a cosine scheulder to a final value of $1e^{-6}$. We use a layer-wise decay of 0.65, and a weight decay of 0.05. We do not use layer scale. We use 8 32GB V100 GPUs for our fine-tuning experiments.

**Linear-probing classification on ImageNet-1k.** For our linear-probing experiments, we mostly follow DINO (Caron et al., 2021b) and DINOv2 (Oquab et al., 2024). We train for 100 epochs and a batch size of 8192 in all our experiments. We use the CLS token only from the last layer of the ViT encoder. Following Oquab et al. (2024), we instantiate multiple linear layers that all take the same encoder output, but are trained with different learning rates for efficiency. This way, the larger backbone inference is done once per iteration, while multiple inferences are done on the much lighter linear layers. We specifically span learning rate values of $\{1e^{-5}, 2e^{-5}, 5e^{-5}, 1e^{-4}, 2e^{-4}, 5e^{-4}, 1e^{-3}, 2e^{-3}, 5e^{-3}, 1e^{-2}, 2e^{-2}, 5e^{-2}, 1e^{-1}\}$, and we linearly scale them using the formula $lr \times batchsize/256$. We report the best performing value. We use a stochastic gradient descent optimizer with the momentum set to 0.9 and no weight decay. We use 8 32GB V100 GPUs for our linear-probing experiments.

**ADE20K semantic segmentation.** In our semantic segmentation evaluations, we also use the iBOT (Zhou et al., 2022) pipeline, which follows BEiT (Bao et al., 2022). Specifically, we use UperNet (Xiao et al., 2018) from the implementation in mmsegmentaion (Contributors, 2020). We fine-tune for 160k iterations using a batch size of 16 and image size of $512 \times 512$. The AdamW optimizer is used with an initial learning

rate of $8e^{-4}$ that is linearly warmed in the first 1500 epochs, then decays to 0 throughout training. We use a layer decay rate of 0.65 and weight decay of 0.05. We use Feature Pyramid Networks (FPNs) of four different scales to modify the feature map sizes generated by the ViT. Specifically, we upsample the output feature of the 4th block and 6th block by 4x and $2\times$, respectively, keep the output from the 8th block the same, and downsample the output feature of the 12th block by $2\times$. For data augmentation, we adopt the default settings in mmsegmentation, which includes random horizontal flipping, random re-scaling with a ratio range of $[0.5, 2.0]$, and random photometric distortion. We use 4 32GB V100 GPUs for our semantic segmentation experiments.

**COCO object detection and instance segmentation.** For object detection and instance segmentation, we use a Cascade Mask R-CNN (Cai & Vasconcelos, 2019; He et al., 2018) implementation with mmdetection (Chen et al., 2019), which produces both bounding boxes and instance masks. We follow the settings in iBOT (Zhou et al., 2022) in that we use multi-scale training with the shorter size between 480 and 800 while the longer size is not larger than 1333. We use a learning rate of $1e^{-4}$, a weight decay of 0.05, and fine-tune for 12 epochs with the learning rate decayed with a rate of 0.1 at epochs 9 and 11. We use a layer decay rate of 0.75 and a batch size of 16. We generate hierarchical feature maps by taking the outputs from layers 4, 6, 8, and 12, and passing them to two deconvolutions, one deconvolution, identity mapping, and max-pooling, respectively. We do not use multi-scale testing. We use 4 32GB V100 GPUs for our object detection experiments.

**Zero-shot classification.** For zero-shot classification, we follow the standard implementation (Radford et al., 2021; Mu et al., 2021) of encoding class labels in a description, such as "a photo of a {label}," and calualting the consine similarity between the text and image embeddings. We consider the highest generated similarity for each image-label pair to be the predicted class. For Harmony, we use the teacher text encoder.

## B  CC12M Zero-shot classification performance

We evaluate zero-shot classification performance of Harmony and CLIP, both trained on the CC12M dataset, across a diverse set of general vision benchmarks. As shown in Table 8, Harmony outperforms CLIP on average.

Table 8: Zero-shot evaluations on CC12M. We compare ViT-B Harmony against CLIP across a range of general and natural image benchmarks for zero-shot classification. Harmony continues to outperform CLIP on average.

|         | Average | Cal101 | STL-10 | Kin700 | MNIST | CLEVR | INet-A | INet-O | INet-R |
|---------|---------|--------|--------|--------|-------|-------|--------|--------|--------|
| CLIP    | 36.6    | 72.8   | 94.8   | 26.6   | **10.7** | **16.2** | 7.9 | **39.5** | 44.1 |
| Harmony | **39.6** | **76.0** | **96.7** | **30.5** | 9.6 | 12.7 | **12.2** | 36.0 | **54.2** |

## C   Reproducing MaskCLIP

As of writing this paper, the code for MaskCLIP's implementation is not yet public so we had to reproduce it ourselves. To do that, we followed the approach in their original paper (Dong et al., 2023). Namely, we define two vision encoders, student $E_V$ and teacher $\bar{E}_V$, and a text encoder $E_T$. We optimize CLIP's contrastive loss, masked self-distillation loss, and masked language modeling simultaneously.

### C.1   Implementation of Masked Self-distillation

We pass a masked image to the student encoder, and the full image to the teacher encoder and optimized the the cross-entropy between masked student patches and unmasked patches from the teacher. More specifically, given an image $x$, we obtain a masked embedding from the student encoder by $e_m = D(E(x_m))$, where $x_m$ is a masked view of $x$ and $D$ is a transformer decoder with 1 layer, 16 attention heads, and with the same embedding dimension as the student encoder $E$. We also obtain a target embedding from the teacher encoder $\bar{E}$ by $e = \bar{E}(x)$. Given a masked vector $m$ with indices corresponding to masked patches, the masked self-distillation loss function then becomes

$$\mathcal{L}_{\text{MaskDist}} = -\frac{1}{\|m\|} \sum_{i=1}^{N} m_i \, \mathcal{H}(P(e), P(e_m)) \, . \tag{14}$$

$P$ is the softmax function, and we used a masking ratio of 75%, removing all mask tokens, just like in MaskCLIP (Dong et al., 2023). For the teacher encoder, we use a momentum parameter of 0.999 that linearly increased to 0.9999 throughout training, and we use minimal augmentation with cropping of scale [0.6, 1] and normalization only. We use the same learning rate parameters in MaskCLIP, and use a loss weight of 0.05 for the masked self-distillation and MLM objectives.

### C.2   Ablation of MaskCLIP

Additionally, we try out different parameters and architectures to further improve the performance of the framework. Namely, we try (1) adding the iBOT head (Zhou et al., 2022), which maps the ViT outputs to vectors of size 8192 using MLP layers, (2) adding the CLS-level objective from iBOT (Zhou et al., 2022) (Our Equation 6), and (3) adjusting the weight for the masked self-distillation and mlm losses. We show our results in Table 9. Having no CLS objective is almost identical to the original MaskCLIP implementation, depending on if centering and sharpening from DINO (Caron et al., 2021b) is done, which is not explicitly described in their paper. Removing the CLS objective and adding the iBOT head result in identical 0-shot performances, both of which are only marginal improvements. We opt to use the iBOT head because it resulted in better generalization across values for our Table 2.

Table 9: MaskCLIP ablations. The "default" settings uses a both the MIM and CLS objectives and does not use the iBOT head. The first column after the default shows the effect of removing the CLS objective, the second column shows the effect of increasing the self-distillation and MLM loss weights from 0.05 to 1, and the third column shows the result of adding the iBOT head.

|  | Default | No CLS obj. | Weighting to 1 | iBOT head | iBOT head & no CLS |
|---|---|---|---|---|---|
| INet-1k 0-shot | 17.3 | 17.7 | 16.7 | 17.7 | 17.4 |

# D  Other Ablations

## D.1  MAE Masking Scheduler

We tried the idea of increasing the masking ratio for pixel reconstruction throughout pre-training using a linear scheduler. This idea was motivated by the fact that having a constant masking ratio is not optimal for scaling the number of iterations, since the task of reconstruction does not become more difficult. This is unlike self-distillation where the teacher model produces better and better representation targets for the students to predict throughout training. As a result, we experimented with using a linear masking ratio scheduler that increases the ratio from 65% to 85% in the first 15 epochs of pre-training. In Table 10 we compare our results to using a static mask ratio of 75%. Classification results mostly stay the same, but segmentation goes down by 1.2%. Further experimentation of the scaling behavior of using a masking scheduler (especially scaling the number of iterations), and tuning of hyper-parameters, like using a lower end value or higher start value, might lead to a different conclusion.

Table 10: Mask scheduler. Classification results stay the same but segmentation goes down by more than 1 percent. The second row linearly scales the masking ratio from 65% to 85 % in the first 15 epochs of pre-training. Both experiments start with the same baseline of CLIP with soft targets + iBOT + MAE.

| Masking Percentage | INET-1K | | | ADE20K |
|---|---|---|---|---|
| | 0-Shot | FT | LIN | SSEG |
| 75% | 20.8 | 79.5 | 64.4 | 36.2 |
| 65% → 85% | 20.6 | 79.6 | 64.5 | 35.0 |

## D.2  CLIP and MAE Augmentations

We compare different data augmentation strategies on CLIP and MAE. For CLIP, we try using the standard augmentation of randomly cropping with a scale in the range [0.4, 1] and random horizontal flipping 50% of the time. We compared this to using the global crop augmentation from DINO and iBOT (Caron et al., 2021b; Zhou et al., 2022), which we described in Section A. We show the comparison in Table 11. Both strategies produce comparable accuracies for classification, but iBOT's augmentation produces higher segmentation IoU.

Furthermore, we explore the effects of using iBOT's global crops with MAE. Specifically, we compare two scenarios: using a standard augmentation as is described above or feeding in both of iBOT's global crops, which is more computationally expensive. We showcase our results in Table 12. Reconstructing both crops substantially improves the fine-tuning performance, so we continue using it in our final Harmony framework.

## D.3  Random and Attentive Masking

We also tried adopting ideas from FLIP (Li et al., 2023) and Attentive Mask CLIP (Yang et al., 2023) in an attempt to improve the efficiency of Harmony. In FLIP (Li et al., 2023), they show that randomly masking

Table 11: CLIP augmentation. Here standard augmentation means random cropping and flipping, with no color distortions or Gaussian blurs. Both start from the same vanilla CLIP baseline.

| Augmentation strategy | INET-1K | | ADE20K |
|---|---|---|---|
| | 0-Shot | FT | SSEG |
| Standard | 17.0 | 76.6 | 33.8 |
| iBOT Global Crop | 16.7 | 76.8 | 35.3 |

Table 12: MAE augmentation. Here standard augmentation means random cropping and flipping, with no color distortions or Gaussian blurs. For global crops, we feed both crops and calculate the final loss as the addition of the MSE from both crops.

| Augmentation strategy | INET-1K (ACC) FT | ADE20K (IoU) SSEG |
|---|---|---|
| Standard | 64.2 | 25.2 |
| iBOT Global Crops | 70.9 | 26.9 |

Table 13: CLIP masking. Here standard augmentation means random cropping and flipping, with no color distortions or Gaussian blurs. For global crops, we feed both crops and calculate the final loss as the addition of the MSE from both crops.

| CLIP masking | INET-1K 0-shot |
|---|---|
| No Mask | 16.7 |
| Attentive (Yang et al., 2023) | 15.9 |
| Random (Li et al., 2023) | 14.9 |

50% of the images being fed to CLIP's image encoder will retain its performance, while substantially improving efficiency. Attentive Mask CLIP builds on this idea by utilizing a ViT's attention matrix (generated using an MAE model, which is our teacher $\bar{E}_V$) to mask only the patches with lowest relationship to the caption description. This is calculated using the CLS token of the vision encoder, which will encode semantic information throughout training because of the nature of CLIP's objective. Both works used significantly larger training datasets than ours, so we wanted to evaluate the ideas on a smaller scale.

We trained a CLIP model with 50% random and attentive masking and present our results in Table 13. Attentive masking performs +1% higher than random masking, but is still worse than no masking. We therefore continued with no masking for our vision-language contrastive learning objective.

## E  Qualitative Results

We show qualitative results for 0-shot classification of Harmony and CLIP in Figure 4. Harmony has a higher chance of identifying the correct guess. Note, that in the first image, Harmony selected a more specific category over the ground truth.

| | Harmony | CLIP |
|---|---|---|
|  | Maypole
Pole
Pinwheel
Torch
Water Tower | Tripod
Hook, Claw
Bow
Mosquito net
Dragonfly |
|  | Goldfinch
Toucan
Goose
Hen
Flamingo | Duck
Hen
Vulture
Toucan
Hummingbird |
|  | Sulphur
Monarch
Lycaenid
Cabbage
Ringlet | Cabbage
Monarch
Admiral
Sulphur
Lycaenid |

Figure 4: Qualitative results of Harmony and CLIP. For 3 different input images, the top 5 guesses of Harmony and CLIP are shown. Ground truth labels are marked in green.

## F  Pseudocode

We present a general pseudocode for our algorithm. More detailed implementation is shared on the GitHub repository. A backward step is performed after each loss to free the computational graph, which saves memory.

---

**Algorithm 1** Harmony Pseudocode

---

    **Input:**
    $E_V, \bar{E}_V$                         // student and teacher vision network
    $E_T, \bar{E}_T$                         // student and teacher text network
    $D_V, D_T$                         // vision and text decoders
    $h_{I_d}, h_{T_d}$                // image and text self-distillation heads
    $m$                            // network momentum
    $\alpha$                           // hard loss weight

1: **for** each *batch* in loader **do**
2:     $i, t = batch$              // get image and text caption
3:     $u, v \leftarrow \mathrm{augment}(i), \mathrm{augment}(i)$       // global views
4:     $\hat{u}, \hat{v} \leftarrow \mathrm{mask}(u), \mathrm{mask}(v)$       // masked image views
5:     $\hat{t} \leftarrow \mathrm{mask}(t)$       // mask text for mlm and self-distillation
6:     $u_s, v_s \leftarrow E_V(u), E_V(v)$   // extract image and text embeddings using student and teacher.
7:     $u_t, v_t \leftarrow \bar{E}_V(u), \bar{E}_V(v)$
8:     $t_s, t_t \leftarrow E_T(t), \bar{E}_T(t)$
9:     $\hat{t}_s, \hat{t}_t \leftarrow E_T(\hat{t}), \bar{E}_T(\hat{t})$
10:
11:     $\mathcal{L} \leftarrow \mathcal{L}_D(h_{I_d}(u_s, v_s, u_t, v_t))$       // distillation loss
12:     $\mathcal{L}$.backward()       // performed at each step to save memory
13:     $\mathcal{L} \leftarrow \alpha \mathcal{L}_{\mathrm{Hard}}(u_s, t_s) + (1 - \alpha)\mathcal{L}_{\mathrm{Soft}}(u_s, t_s, u_t, t_t)$   // soft and hard contrastive loss
14:     $\mathcal{L}$.backward()
15:     $\mathcal{L} \leftarrow \mathcal{L}_R(D_V(u_s, v_s), i)$       // pixel reconstruction loss
16:     $\mathcal{L}$.backward()
17:     $\mathcal{L} \leftarrow \mathcal{L}_M(D_T(\hat{t}_s), t)$       // masked language modeling loss
18:     $\mathcal{L}$.backward()
19:     $\mathcal{L} \leftarrow \mathcal{L}_{TD}(h_{T_d}(\hat{t}_s), h_{T_d}(t_t))$       // text-distillation loss
20:     $\mathcal{L}$.backward()
21:
22:     **update**$(E_V)$       // backpropagation based on the above losses
23:     **update**$(E_T)$
24:     $\bar{E}_V$.params $\leftarrow m\,\bar{E}_V$.params $+ (1 - m), E_V$.params       // EMA teacher update
25:     $\bar{E}_T$.params $\leftarrow m\,\bar{E}_T$.params $+ (1 - m), E_T$.params
26: **end for**

---

## G   Comparison with DINOv2

DINOv2 (Oquab et al., 2024) is an obvious comparison with Harmony, as it is an improved version of iBOT (Zhou et al., 2022). However, we do not include it in our results, because we observed poor performance when pre-training DINOv2 on CC3M (linear probing accuracy below 25%), using the publicly shared codebase. Given that the modifications introduced by DINOv2 beyond iBOT are relatively minor, with many being just hyperparameter changes (see Table 1 in Oquab et al. (2024)), we would expect similar performance. We suspect a bug may be present in the code, though we were unable to identify it. As a result, we felt it would be more fair to exclude this comparison.

