# OpenReview forum: "Harmony: A Joint Self-Supervised and Weakly-Supervised Framework for Learning General Purpose Visual Representations"
_TMLR — Accepted by TMLR_

### Review · Reviewer_oog6 · 2025-04-02

**Summary Of Contributions:**

Harmony proposes a framework for learning general-purpose visual representations by combining self-supervised and weakly-supervised learning. Instead of using hard targets for aligning text-image embeddings in a shared space, Harmony introduces soft targets to improve alignment. Additionally, it enhances visual representation learning by integrating both generative and discriminative self-supervised learning techniques into the image encoder.

**Audience:**

Yes

**Claims And Evidence:**

Yes

**Requested Changes:**

See the weakness section.

**Strengths And Weaknesses:**

**Strengths:**

1-The paper provides a thorough and well-structured review of relevant prior work.

2-The problem addressed in this work is of critical importance, garnering substantial attention in computer vision research due to its broad practical applications in tasks such as cross-modal retrieval, image classification, and downstream vision systems.

3-The authors conducted extensive experiments on multiple benchmark datasets, demonstrating that their approach outperforms existing methods

**Weaknesses:**

1-In contrastive learning, such as MoCo-v2 and SimCLR, the negative example plays a crucial role where positive pairs are attracted, and negative pairs are repelled in the embedding space. In this paper, particularly in the abstract and contribution, the authors state that their approach does not use negative examples. However, in different part of the paper the authors state that they are using soft contrastive learning. On page 5, paragraph 2, they describe that they are maximising the similarity between paired embedding sets, $v_i$ and $t_j$, where i = j and minimise the similarity between unpaired sets where $i \neq j$.  which means that they are using negative examples to train the models. Also, Equations 2, 3, 4, and 5 which are used to align text and images in the shared space, demonstrating that they employ a contrastive approach, which relies on negative samples.  **Could the author clarify this confusion?**

2- The notation $t_k$ in the denominator of equation 2 does not clearly refer to negative examples or what? **Could the author clarify what this notation refers to?**

3-In Section 3.4, it is stated that the soft target is explained in Section 5 (Conclusion), where it is actually located in Section 3.

4- The authors state in section 3.4: “Even though the goal of this work is visual representation learning, we hypothesise that improving the textual representations in our text encoder$ E_T$ can improve visual representation in the vision encoder $E_V$ both by making the soft learning targets described in Section 5 more meaningful and by improving the overall contrastive learning task”. **The statement requires further clarification on how the text encoder enhances visual representation in the vision encoder and how it improves the contrastive learning task.**

5- The authors refer to Table 3 to support their claim about the text encoder.  In Table 3, they compare their approach with the baseline (i.e., CLIP), which does not have discriminative and generative parts (i.e., self-supervised learning) for training the image encoder. Thus, it is unclear whether this gain in performance is due to SSL or to the text encoder with MLM. **To make a fair comparison and prove that the addition part (i.e., EMA with MLM) caused this improvement, they should compare their approach with and without this part to see the effect of this part.**

6- **We encourage the authors to incorporate pseudo-code for the algorithm in the paper, similar to the counterparts, to clarify any ambiguity in the approach.**

---

> ### Author Response · Authors · 2025-04-21
>
> We thank the reviewer for their thoughtful comments.
> Addressing the weaknesses:
>
> (1)
> In the abstract and contribution, we were specifically talking about negative examples in the self-supervised learning path (using iBOT which does not use negative examples, compared to SLIP which uses SimCLR which has negative examples). We now changed the abstract and contribution to "negative examples in the self-supervised learning path" to be more specific. In the CLIP-path, when we say negative examples, we mean hard negatives, where the target is set to 0 (representing no alignment at all). Our method **does use** hard negative examples (as can be seen in equation 5), but the contribution from hard negative targets in the loss decreases slowly throughout training (controlled by alpha in equation 5, where alpha slowly decreases throughout training). Equation 2 is the softmax on the similarity between the image and text embeddings in the batch. Specifically, you can see in equations 3 and 4, that, unlike the original clip loss (equation 3) our soft loss does not assign hard negative targets (encoded in the identity matrix of equation 3) for non-paired examples, and instead uses the EMA models to generate those "soft" targets, which have values [0,1].
>
> (2)
> We don't consider this to be a negative example, as here we are just calculating the cosine similarity between an example *i* and other non-paired examples in the batch. They would only be **negative** based on our optimization, where if we set their target to 0 they would be considered hard negatives. In the optimization in our soft loss, examples that are not paired with *i* are not assigned target 0, and instead we use the EMA models to generate their targets.
>
> (3)
> Fixed, thanks.
>
> (4/5)
> We make the hypothesis, then support it in Table 3. Specifically, in Table 3 we are encouraging the readers to look at the first row and compare it to the row after adding the text-distillation losses (row before last), where we see that the fine-tuning accuracy increases by 0.4\% after adding the text-losses only. However, we understand that this might not be conclusive enough, so we opted to remove this statement instead.
>
> (6) We added pseudocode on the last page in the appendix. The pseudocode is very general, as we will share the GitHub repo for all the code after acceptance.

---

### Review · Reviewer_XaFn · 2025-04-08

**Summary Of Contributions:**

This work proposes Harmony, a self-supervised and weakly supervised framework for training a visual encoder. Harmony builds on a line of work which have extended CLIP, which is here positioned as Weakly-supervised learning or text-guided learning, with additional self-supervised losses. Prior work SLIP adds the SimCLR loss to CLIP, and MaskCLIP adds self-distillation and masked image modelling losses.

In this paper this idea is extended even further, by (1) adapting the contrastive loss to use soft-targets for the text-guidance, (2) visual feature self-distillation (local and global), (3) MSE loss pixel reconstruction, (4) a masked language modelling loss, and (5) text self-distillation.

Experimental results show that this approach leads to a stronger visual encoder across a variety of vision tasks.

**Audience:**

Yes

**Claims And Evidence:**

Yes

**Requested Changes:**

W1: While the work as is seems fairly solid, leading to clear improvements (despite some computational cost gain during training) on vision tasks, it does (for the lack of a better word) seem a little dated. This area of research is very active, with many new works coming out, and yet most cited work is from before 2024 (with DINOv2 being the only reference from 2024).

This would not have to be an obstacle, but it is reinforced by the issue that the works which are compared against are all pre-2023, with newer versions of these models having being released since. For instance, SigLIP (for which the paper is never cited in this manuscript) in Table 1 has since been improved upon by SigLIP2 [A].

Whilst SigLIP2 is very recent (February 2025), there are other works that are similar to the proposed paper that would be meaningful to compare against. Like SILC [B] and Brave [C]. Moreover, while DINOv2 is mentioned in the paper it is not compared against. Since the main contribution of this work is driven by performance it is necessary to compare against recent work.

[A] SigLIP 2: Multilingual Vision-Language Encoders with Improved Semantic Understanding, Localization, and Dense Features. Tschannen et al. 2025.
[B] SILC: Improving Vision Language Pretraining with Self-Distillation. Naeem et al. ICLR 2024.
[C] BRAVE: Broadening the visual encoding of vision-language models. Kar et al. ECCV 2024

W2: This work builds on CLIP, which is a cross-modal approach that jointly learns a vision and a text encoder, and similarly many of the works that build on it similarly learn an encoder for both modalities, as is done for Harmony. Yet, the emphasis of the paper and the experiments is on vision tasks. Comparable works typically also evaluate on cross-modal tasks, such as image-to-text and text-to-image retrieval, and from the current paper it is not obvious how Harmony performs on cross-modal tasks. This may be by choice, but it would strengthen the paper if it was more clear why it is positioned and evaluated as a vision-first model, rather than a vision-language model.

Moreover, it is not made explicit in the paper but presumably the 0-shot experiments are done in CLIP style, i.e., by encoding the class labels with the text encoder and then argmax over cosine distances between encoded class labels and visual features?

**Strengths And Weaknesses:**

The problem is clearly introduced, and the paper transparently lays out the connection between prior work and how this work uses components from prior work to construct the Harmony model. The experimental details are clear and experiments are performed across a variety of datasets showing the gains of Harmony. Moreover, the ablations are meaningful and there are results which make clear the computational costs.

---

> ### Author Response · Authors · 2025-04-21
>
> We thank the reviewer for their thoughtful comments. Addressing the weaknesses:
>
> (1)
>
> We agree with the reviewer, and have added two newer methods: SLIC as suggested (ECCV, 2024) and DetailCLIP (ICLR 2025). We are adding results to the table as we get them. See the updated Tables 1, 2, and 3. Figure 1 will also be updated once we have all the results. For BRAVE, the other method you suggested, it combines the strengths of many already trained vision encoders, which is somewhat orthogonal to our work, since we focus on pre-training methods instead. DINOv2 is another obvious comparison. However, we were not able to get reasonable results with pre-training DINOv2 models from scratch, even after spending weeks trying to find and fix any potential issues in the pre-training and evaluation pipelines. Technically DINOv2 is just iBOT, with little changes to the objective (see Table 1 in Oquab, Maxime, et al. "Dinov2: Learning robust visual features without supervision." arXiv preprint arXiv:2304.07193 (2023)). Other modifications include changing the batch size, patch size, activation functions, and other hyperparameters. Therefore, it was hard to justify the large gap in performance between iBOT and DINOv2 that we were observing (DINOv2 getting <25\% accuracy in linear probing), so we opted not to include it since we think there likely is an issue somewhere. For SigLIP2, the training codebase is not shared publicly, and there is currently no public implementation like OpenCLIP or elsewhere (as far as we are aware). See https://github.com/mlfoundations/open_clip/issues/1041.
>
> (2)
>
> Actually, we also reported image-to-text and text-to-image retrieval results as well, but it was in the appendix. We have now moved it to the main paper (Table 3). This with the zero-shot classification performance should be enough to convince readers that the Harmony framework is good at multimodal tasks.
>
> "Moreover, it is not made explicit in the paper but presumably the 0-shot experiments are done in CLIP style, i.e., by encoding the class labels with the text encoder and then argmax over cosine distances between encoded class labels and visual features?"
>
> That's correct. I think this is the standard way to do it, so we don't need to include it in our paper.

---

> > ### Comment · Reviewer_XaFn · 2025-04-22
> >
> > Thank you for the additional experiments and clarification.
> >
> > > we opted not to include it [DINOv2] since we think there likely is an issue somewhere
> >
> > While it is a pity not to have the comparison, this decision makes sense. A brief mention in the paper of this issue does seem necessary.
> >
> > > have added two newer methods:  SLIC as suggested (ECCV, 2024) and DetailCLIP (ICLR 2025)
> >
> > It does seem necessary to update the text to describe these new results - as they (and SLIC in particular) appear to be competitive with the proposed Harmony model. For Table 1 and 2 SILC appears to perform on par with Harmony - given the small performance difference it would be necessary to report average performance (with standard deviation) over multiple random seeds to conclude which performs better. Otherwise it would be more apt to state they perform on par.
> >
> > > we also reported image-to-text and text-to-image retrieval results
> >
> > Thank you for moving these to the main paper; this indeed better supports the multimodal aspect.
> >
> > > I think this is the standard way to do it, so we don't need to include it in our paper.
> >
> > Since there is already an appendix with experimental details it would make sense to briefly document this, even if just to confirm that this work followed the standard approach.
> >
> > > which is somewhat orthogonal to our work, since we focus on pre-training methods instead
> >
> > While I understand the distinction, this does further highlight a limitation of the current work. Which is also pointed out by reviewer S5KU - currently all pretraining is done on the smallest pretraining scale (CC3M). This indeed results in low downstream scores across all methods.
> >
> > I echo reviewer S5KU's request for further analysis on data scaling. Alternatively, it may be worth to explore a framing of the paper as pre-training in a data constrained setting.

---

> > > ### Author Response · Authors · 2025-04-22
> > >
> > > Thank you for your thoughtful comments.
> > >
> > > > While it is a pity not to have the comparison, this decision makes sense. A brief mention in the paper of this issue does seem necessary.
> > >
> > > We now mention this in the appendix (section C).
> > >
> > > > It does seem necessary to update the text to describe these new results - as they (and SLIC in particular) appear to be competitive with the proposed Harmony model. For Table 1 and 2 SILC appears to perform on par with Harmony
> > >
> > > We have updated the abstract to highlight the fact that Harmony outperforms SILC only on detection, fine-tuning and linear probing classification. We also changed the contributions to say that our method is **superior or competitive** to leading methods. We also updated the results section (4.2), and conclusion to reflect this.
> > >
> > > > given the small performance difference it would be necessary to report average performance (with standard deviation) over multiple random seeds to conclude which performs better. Otherwise it would be more apt to state they perform on par.
> > >
> > > For this, running the experiments with different seeds would be too computationally heavy right now. Each downstream evaluation takes about 1-1.5 days to run on 8 V100 GPUs, and we have access to only 16-24 at a time. It would take more than 3 weeks of run time to do this for all methods. We could technically do this only for Harmony and SILC, but then it's not obvious why we would need to do that; all methods are run on the exact same setup with the same seed. It is also not obvious that we would observe significantly different results using different seeds. This is also the current standard of reporting downstream performance. For table 2, the task is deterministic, so we should not observe different performance with different seeds.
> > >
> > > > Since there is already an appendix with experimental details it would make sense to briefly document this, even if just to confirm that this work followed the standard approach.
> > >
> > > We now mention this in the appendix.
> > >
> > > > While I understand the distinction, this does further highlight a limitation of the current work. Which is also pointed out by reviewer S5KU - currently all pretraining is done on the smallest pretraining scale (CC3M). This indeed results in low downstream scores across all methods. I echo reviewer S5KU's request for further analysis on data scaling.
> > >
> > > See the general response on this.
> > >
> > > > Alternatively, it may be worth to explore a framing of the paper as pre-training in a data constrained setting.
> > >
> > > We now mention this in the abstract and contribution, where we say that Harmony is particularly suited for data-constrained settings because we optimize for five objective simultaneously, which means that we more effectively utilize each single data example.

---

> > > > ### Comment · Reviewer_XaFn · 2025-04-23
> > > >
> > > > > it's not obvious why we would need to do that; all methods are run on the exact same setup with the same seed. It is also not obvious that we would observe significantly different results using different seeds.
> > > >
> > > > A particular random seed may benefit one approach over another as it changes random initialisation and the order of the training data - which will lead to variation in downstream performance. Unfortunately, it is not completely uncommon for works to 'optimse' the seed. See also David Picard's paper 'Torch.manual_seed(3407) is all you need: On the influence of random seeds in deep learning architectures for computer vision'.
> > > >
> > > > As such averaging over multiple runs initialised with (randomly selected) different seeds is a standard approach to get better estimates of actual model performance. This would provide a stronger statistical foundation to make claims about improved model performance, especially in cases where the gap is <1 %-point.
> > > >
> > > > However, given the computational cost of running such additional experiments and the adjusted phrasing to state that Harmony is competitve with other approaches I believe this point is sufficiently addressed.
> > > >
> > > > I have no further concerns.

---

### Review · Reviewer_S5KU · 2025-04-13

**Summary Of Contributions:**

This paper proposes a hybrid architecture for pretraining with self-supervision and language supervision to enhance the dense prediction performance on downstream datasets, which incorporates five loss functions: (1) vision student-teacher (ST) distillation loss; (2) language ST distillation loss; (3) visual masked modeling loss; (4) language masked modeling loss; and (5) vision-language contrastive ST distillation loss.

Experimental results on IN1K, ADE20K, and COCO validate the effectiveness of the proposed architecture. Besides, better scaling results are demonstrated across the different model capacities.

**Audience:**

Yes

**Broader Impact Concerns:**

If the author wants to draw solid conclusions for the pretraining community, more scaled experiments are required, e.g., CC12M.

**Claims And Evidence:**

Yes

**Requested Changes:**

See weaknesses.

**Strengths And Weaknesses:**

Strengths:

- The motivation is clear and the writing is easy to follow.
- The architecture is sophisticated and requires some engineering tuning.

Weaknesses:

- Pretraining on CC3M may not be scaled enough. It would be better if authors could include ablations on data scaling. In recent years, the pretraining scale has been ascended to scenarios where CC3M is considered the smallest pretraining scale [1]. Even for early work IBOT, they include pretraining on IN21K. Although it is not friendly to most labs, it may be an important issue for work in this direction. As it is witnessed that the zero-shot performance on the downstream datasets is quite low, the larger-scale dataset for vision-language pretraining is required.
- Writing issue on related work: need to mention the differences of this work compared with prior works at each subsection to highlight technical differences/contributions to enhance readability.
- Weakly supervised learning might be a broad term that can be narrowed down to language-supervised pretraining or something else.
- Notation issue: the function $P$ in eq. 2 and 3 take different inputs.

[1] No “Zero-Shot” Without Exponential Data: Pretraining Concept Frequency Determines Multimodal Model Performance

---

> ### Author Response · Authors · 2025-04-22
>
> We thank the reviewer for their thoughtful comments. Addressing the weaknesses:
>
> > Pretraining on CC3M may not be scaled enough. It would be better if authors could include ablations on data scaling. In recent years, the pretraining scale has been ascended to scenarios where CC3M is considered the smallest pretraining scale [1]. Even for early work IBOT, they include pretraining on IN21K. Although it is not friendly to most labs, it may be an important issue for work in this direction. As it is witnessed that the zero-shot performance on the downstream datasets is quite low, the larger-scale dataset for vision-language pretraining is required.
>
> See our general response about this
>
> > Writing issue on related work: need to mention the differences of this work compared with prior works at each subsection to highlight technical differences/contributions to enhance readability.
>
> We now add this in the last subsection of the related works.
>
> > Weakly supervised learning might be a broad term that can be narrowed down to language-supervised pretraining or something else.
>
> We agree and have changed it to language-supervised or language guided learning where appropriate, but still kept the reference to weakly-supervised learning, as it still belongs to this broader category.
>
> > Notation issue: the function P in eq. 2 and 3 take different inputs.
>
> Equation 3 shows $P$ which is the vectorized version of $P_v$ in equation 2. We now explain that in the text.

---

> > ### Comment · Reviewer_S5KU · 2025-04-23
> >
> > Thanks for all your efforts. It is highly suggested that, if computational resources allow, you include results on a more scaled version, not necessarily the CC12M, which we acknowledge is costly to train. Instead, you can run experiments on the randomly sampled version of CC12M, e.g., with a 2x CC3M scale, to demonstrate the general scaling trend of your method and baselines before the rebuttal deadline.

---

> > > ### Comment · Reviewer_S5KU · 2025-04-23
> > >
> > > My remaining concerns have been addressed.

---

### Author Response · Authors · 2025-04-22

## On data scaling

Reviewers **S5KU** and **XaFn** have requested further experimentations on a scaled dataset like YFCC15M or CC12M. Both datasets are 4-5x larger than the current dataset we are using (CC3M), and given that we are computationally restricted, adding data scalled experiments for all methods is beyond our limits. Instead, we could add results with pre-training **CLIP vs Harmony only**, likely using a ViT-S on YFCC15M, but that will also require more time, and we will likely need more time beyond the May 11 deadline, although we will try.

The reviewers have requested the addition of more scalled datasets, mainly because of the low zero shot performance. This is fair. However, all methods are trained on the same data, and here we are trying to highlight the performance differences between the methods, and not absolute performance. While larger data would help draw stronger conclusions, we suspect that it won't show completely different trends in terms of results.

Alternately, thanks to reviewer **XaFn's** suggestion, we now also slightly frame this work as more of pre-training in a data constrained setting (see updated abstracts and contributions). From the abstract, "Harmony optimizes for five different objectives simultaneously, efficiently utilizing the supervision in each data example, making it even more suited in data-constrained settings."

We suggest that the reviewers consider this rebuttal, but otherwise we will run the larger data scalled experiments on CLIP and Harmony and add results whenever available.

## Comparison with more recent methods and contribution implications

Per **XaFn's**, we now include two new comparisons, SILC and DetailCLIP (see updates tables 1, 2, and 3). Given that SILC performs similarly to Harmony, we now describe our method as being **superior or competitive** with other leading methods.

Furthermore, we now add **another contribution to our paper** (see the 4th contribtuion), which is that we open-source our code that is modular (meaning you can combine any two or more methods, MAE + CLIP, DINO + CLIP, DINO + MAE, etc.) and that includes implementation of methods that were previously not open-sourced (SILC and MaskCLIP).

We thank the reviewers for their comments and hope that the current version of the paper is much better.

---

> ### Author Response · Authors · 2025-04-30
>
> **Update**
>
> We are trying to launch training runs for CC12M, but it is taking a long time to get compute on our cluster, since we are requesting more GPUs & time limit. We are planning to update our paper with results from CC12M, but we can not do that before the May 11th deadline.
>
> However, we still would like to update the paper with the CC12M results once we have them, even after the deadline, if possible.

---

### Decision · Action_Editor_w45z · 2025-05-16

**Recommendation:** Accept with minor revision

**Comment:**

The initial reviews raised concerns about the scale of pretraining (especially given the low zero-shot performance), outdated comparison methods, and a need to clarify several aspects of the proposed work.

These issues have been satisfactorily addressed in the rebuttal and the revised manuscript. For the final paper, the authors are required to add results on a larger dataset like YFCC15M as discussed in the rebuttal.

**Audience:**

Although the contributions of this work may be considered somewhat incremental, the TMLR audience should nevertheless have some interest.

**Claims And Evidence:**

Sufficiently convincing evidence is provided to support the claims, particularly after more recent comparison methods were added during the discussion period. The three reviewers and the AE all agree that the claims are adequately substantiated.

---

> ### Author Response · Authors · 2025-06-04
>
> We thank the Action Editor and reviewers for their helpful feedback.
>
> We submitted the camera ready version. Results for the bigger dataset (CC12M) are now available on Figure 3 and Table 8 in the Appendix.